# Novel inhibitors of microtubule organization and phragmoplast formation in diverse plant species

Yusuke Kimata[1] , Moé Yamada[2] , Takashi Murata[3] , Keiko Kuwata[4] , Ayato Sato[4] , Takamasa Suzuki[5] , Daisuke Kurihara[4,6] , Mitsuyasu Hasebe[7,8] , Tetsuya Higashiyama[4,9] , Minako Ueda[1,10]

Cell division is essential for development and involves spindle assembly, chromosome separation, and cytokinesis. In plants, the genetic tools for controlling the events in cell division at the desired time are limited and ineffective owing to high redundancy and lethality. Therefore, we screened cell division–affecting compounds in *Arabidopsis thaliana* zygotes, whose cell division is traceable without time-lapse observations. We then determined the target events of the identified compounds using live-cell imaging of tobacco BY-2 cells. Subsequently, we isolated two compounds, PD-180970 and PP2, neither of which caused lethal damage. PD-180970 disrupted microtubule (MT) organization and, thus, nuclear separation, and PP2 blocked phragmoplast formation and impaired cytokinesis. Phosphoproteomic analysis showed that these compounds reduced the phosphorylation of diverse proteins, including MT-associated proteins (MAP70) and class II Kinesin-12. Moreover, these compounds were effective in multiple plant species, such as cucumber (*Cucumis sativus*) and moss (*Physcomitrium patens*). These properties make PD-180970 and PP2 useful tools for transiently controlling plant cell division at key manipulation nodes conserved across diverse plant species.

## Introduction

In multicellular organisms, growth and pattern formation depend on accurate cell division. Plant cell division involves intracellular events such as chromosome condensation and spindle assembly for nuclear division and the formation of a preprophase band (PPB) and phragmoplast for cytokinesis (Müller, 2019). Manipulating specific events at the desired time is crucial for controlling development and analyzing the cell division machinery. However,

developing such tools is still challenging in plants because genetic manipulations do not work effectively owing to a high gene redundancy, mutant lethality, and various pleiotropic effects. For example, in *Arabidopsis thaliana* (Arabidopsis), multiple mutations in related cell division regulators cause gametophytic or embryonic lethality, whereas single mutants display no or minor defects (Pan et al, 2004; Lee et al, 2007). In addition, mutants that fail to regulate microtubules (MT), a pivotal component of the spindle, PPB, and phragmoplasts, inhibit cell division and exhibit pleiotropic phenotypes such as those affecting stress responses and tissue specification (Luptovčiak et al, 2017).

To overcome redundancy, tools such as chemical inhibitors that bind to the conserved domains of homologous proteins are required to impede the functions of related regulators (Serrano et al, 2015). Furthermore, chemical inhibitors can be applied at any time and therefore represent powerful tools that can ensure transient and stage-specific control and avoid lethality and side effects. The use of various specific inhibitors in animal studies has greatly advanced our understanding of cell division mechanisms (Ong & Torres, 2019).

In plants, chemical effects on cell division control can be adequately monitored using time-lapse observations of the *Nicotiana tabacum* (tobacco) Bright Yellow-2 (BY-2) cell strain (Nagata et al, 1992). BY-2 suspension cultures are suitable for performing intracellular and biochemical analyses such as high-speed live imaging of phragmoplast formation and purifying phosphorylated proteins from synchronized cells (Sasabe et al, 2011; Murata et al, 2013). However, in suspension cultures, cells at different division stages are mixed, and their positions in liquid media constantly change. Therefore, skilled professionals and specific devices for time-lapse observations are needed to trace individual cell states and determine the exact effects of the tested compounds on cell division.

Arabidopsis zygotes are single cells with thoroughly documented cell division time points (time courses) and patterns that can be easily identified in the individual ovules (Mansfield & Briarty, 1991;

[1]Graduate School of Life Sciences, Tohoku University, Sendai, Japan    [2]Department of Biological Science, Division of Natural Science, Graduate School of Science, Nagoya University, Nagoya, Japan    [3]Department of Applied Bioscience, Kanagawa Institute of Technology, Atsugi, Japan    [4]Institute of Transformative Bio-Molecules (WPI-ITbM), Nagoya University, Nagoya, Japan    [5]College of Bioscience and Biotechnology, Chubu University, Kasugai, Japan    [6]Institute for Advanced Research (IAR), Nagoya University, Nagoya, Japan    [7]National Institute for Basic Biology, Okazaki, Japan    [8]School of Life Science, The Graduate University for Advanced Studies, Okazaki, Japan    [9]Department of Biological Sciences, Graduate School of Science, The University of Tokyo, Tokyo, Japan    [10]Suntory Rising Stars Encouragement Program in Life Sciences (SunRiSE), Kyoto, Japan

Correspondence: minako.ueda.e7@tohoku.ac.jp

Ueda et al, 2017). For example, it takes ~20 h after fertilization before the zygote undergoes its first cell division. The average time required for subsequent cell divisions is 7–9 h until the early globular embryo is formed (Gooh et al, 2015). The zygote's distinct anatomy and regular cell division durations make it an ideal platform to investigate cell division events and identify any morphological deviation. Moreover, a reliable ovule cultivation system for time-lapse observations has been established to record the developmental time course of growing zygotes (Gooh et al, 2015; Kurihara et al, 2017; Ueda et al, 2020). Using high-resolution microscopy, the ovule cultivation system can be applied to evaluate pharmacological effects on zygotic division and monitor intracellular dynamics (Kimata et al, 2016, 2020; Matsumoto et al, 2021).

In the present study, we introduced an ovule cultivation system to screen compounds that affect zygotic cell division. The effects of the identified compounds on cell division were assessed using time-lapse observations of tobacco BY-2 cells. Using mass spectrometry (MS)–based protein identification and assessments using various cell types and plant species, we identified two plant cell division inhibitors, PD-180970 and PP2, which block MT organization and phragmoplast formation, respectively, in diverse plant species.

# Results

## Identifying potent plant cell division inhibitors based on chemical screening of Arabidopsis zygote

We established an in vitro ovule cultivation system for chemical screening. To observe cell division, we used a dual-color marker that simultaneously labels the embryonic nuclei and plasma membranes (histone/PM) (Gooh et al, 2015). Self-pollinated and zygote-containing ovules were isolated from the pistils and incubated in culture media for 2 d (Fig 1A). When cultivated with a control solvent (dimethyl sulfoxide, DMSO), all fluorescent-positive (living) embryos developed into the globular stage without morphological defects (Fig 1B, DMSO), as indicated by the ratio of arrested to abnormal embryos (0%, n = 132). When cultivated with a known MT inhibitor, oryzalin, most zygotes did not divide (93%, n = 28; Fig 1B, oryzalin). Therefore, we concluded that this ovule cultivation system could examine the inhibitory effects of the applied compounds and subsequently screen for potential plant cell division inhibitors.

Two commercially available chemical libraries were selected for screening. The Library Of Pharmacologically Active Compounds (LOPAC) Pfizer library (Sigma-Aldrich) consists of 90 bioactive compounds, whose targets have been already identified in animals. The SCREEN-WELL Kinase Inhibitor library (Enzo Life Sciences) includes 80 reagents, each inhibiting specific kinases in animals. Individual compounds were applied to cultivated ovules at a concentration of 10 $\mu$M because both libraries were presolubilized at 10 mM in DMSO, and DMSO concentration exceeding 0.1% harms cell division in this cultivation system. Three antiproliferative compounds were identified with two technical replications: PD-180970 and 5-iodotubercidin (5-ITu) from LOPAC Pfizer and PP2 from the SCREEN-WELL Kinase Inhibitor (Fig 1B). Zygotes cultivated with

PD-180970 (100%, n = 50) and PP2 (100%, n = 78) showed severe cell division arrest similar to those cultivated with oryzalin (Fig 1B). Moreover, PD-180970 and PP2 were effective at concentrations of 10 nM and 1 $\mu$M, respectively (Fig 1C). In contrast, 5-ITu only partially inhibited cell division and resulted in abnormal embryos with fewer cells compared with the DMSO-treated embryos (77%, n = 35; Fig 1B). The 5-ITu compound reportedly inhibits Haspin kinase in both animals and plants and causes chromosome misalignment during mitosis in tobacco BY-2 cells (De Antoni et al, 2012; Kozgunova et al, 2016). In contrast, PD-180970 and PP2 inhibit specific tyrosine kinases, Bcr-Abl and Src, respectively, in animals (Hanke et al, 1996; Dorsey et al, 2000). However, their targets in plants were not predicted because these kinases have no homologs. Therefore, we focused on PD-180970 and PP2 for further analysis.

## PD-180970 and PP2 inhibit MT organization and phragmoplast formation, respectively

To obtain additional insights into the inhibitory mechanisms of PD-180970 and PP2, we examined the cellular dynamics of cultivated zygotes using a time-lapse observation system (Fig 1D–G and Video 1). In DMSO-treated zygotes, the zygotic nucleus was condensed and then separated. This was followed by cell plate formation, which is indicative of cytokinesis, resulting in two decondensed nuclei in separate cells (Fig 1D). In oryzalin- and PD-180970–treated zygotes, the nucleus failed to separate, and no cell plates were formed, resulting in one decondensed nucleus per cell (Fig 1E and F). In PP2-treated zygotes, the zygotic nucleus condensed and separated normally; however, no cell plate was formed, resulting in two accompanying nuclei or one fused nucleus in a cell (Fig 1G).

Because MT pivotally influences mitosis and cytokinesis, we performed time-lapse observations to examine the effects of PD-180970 and PP2 on MT markers in zygotes using two-photon excitation microscopy (Fig 1H–K and Video 2). The PPB, spindle, phragmoplast, and cortical MT arrays were observed in DMSO-treated zygotes, as shown previously (Fig 1H) (Kimata et al, 2016). These MT structures were disrupted in the oryzalin- and PD-180970–treated zygotes (Fig 1I and J). In contrast, only phragmoplasts were lost in the PP2-treated zygotes (Fig 1K). In addition, none of the compounds disrupted the longitudinal alignment of the other cytoskeleton, actin filaments (F-actin), after 1 h of treatment (Fig S1A), whereas the same conditions caused a disruption of the F-actin array in the presence of an F-actin–specific inhibitor (Kimata et al, 2016). These results suggest that PD-180970 and PP2 specifically affect MT alignment and phragmoplast formation, respectively.

To further assess whether the inhibitory effects were specific to Arabidopsis zygotes or general to other plants, we performed time-lapse observations of the BY–GTRC strain, wherein MT and centromeric histones were labeled in tobacco BY-2 cells (MT/centromere; Fig S1B and Video 3) (Kurihara et al, 2008). The same effects observed in Arabidopsis zygotes were also observed in tobacco BY-2 cells; PD-180970 and oryzalin inhibited nuclear division, and PP2 blocked phragmoplast formation (Fig S1B and Video 3). Therefore, we concluded that these compounds targeted essential cell division regulators common in Arabidopsis zygotes and tobacco suspension–cultured cells.

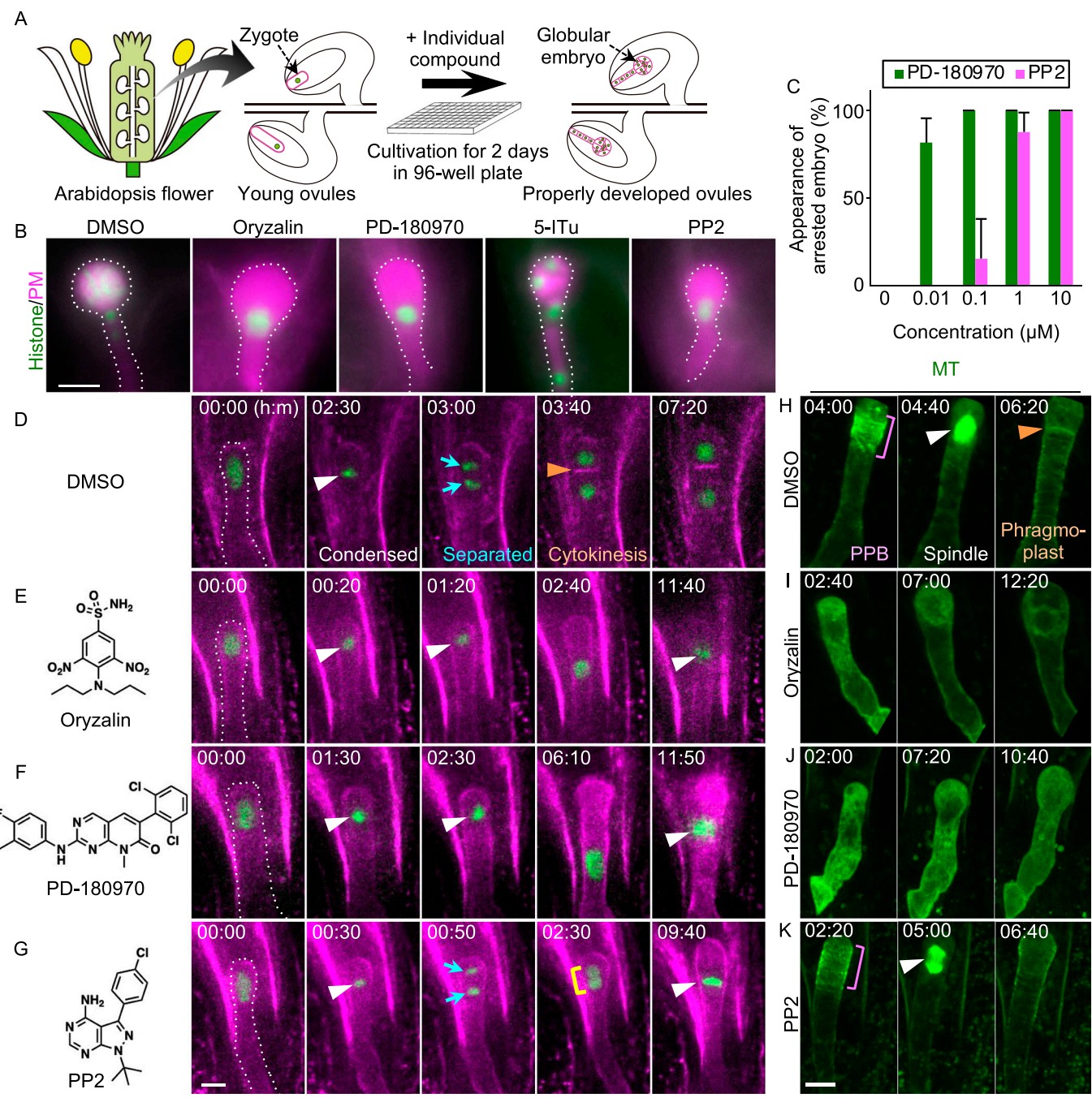

Figure 1. Chemical screening based on Arabidopsis zygote to identify cell division inhibitors.
(A) Schematic procedure of chemical screening using in vitro ovule cultivation system. (B) Epi-fluorescent images of the embryos expressing histone/PM marker at 2 d after the application of indicated compounds. Embryos are outlined by dotted lines. (C) Dosage-dependent inhibition of Arabidopsis zygote over the concentrations of PD-180970 and PP2. Appearance of arrested embryos is the ratio of embryos with fewer cells compared with globular embryos among total living embryos at 2 d after the incubation. Error bars represent the SD (n = 3). More than 52 ovules were counted for each test. (D, E, F, G) Time-lapse observation of the zygote expressing the histone/PM marker in the presence of the indicated compounds. The left drawings show the structures of respective compounds. Numbers indicate the time (h:min) from the first frame, and the starting zygotes are outlined by dotted lines. White and orange arrowheads indicate the spindle and newly formed cell plate, respectively. Cyan arrows and yellow rectangle show separated nuclei and two accompanied nuclei, respectively. (H, I, J, K) Time-lapse observation of MT alignment in Arabidopsis zygote in the presence of indicated compounds. Maximum-intensity projection images are shown, and numbers indicate the time (h:min) from the first frame. Magenta rectangles mark PPB formation in dividing zygotes. White and orange arrowheads indicate the spindle and phragmoplast, respectively. (B, D, E, F, G, H, I, J, K) Scale bars: 30 μm (B) and 10 μm (D, E, F, G, H, I, J, K).

**PD-180970 and PP2 do not irreversibly damage the viability**

We examined the long-term effects and toxicity of PD-180970 and PP2 to test their utility as cell division inhibitors in physiological experiments. We used Arabidopsis seedlings, which can be easily transferred between inhibitor-free and inhibitor-containing media, with root meristems showing active and regular cell division (Fig S2) (Benfey & Schiefelbein, 1994). After a 1-d treatment with the compounds, the root tips of the histone marker plants were stained with propidium iodide (PI) to visualize the plasma membrane (histone + PI; Fig S2A). Compared with DMSO-treated seedlings, oryzalin-treated seedlings had thicker roots consisting of enlarged nuclei (Fig S2A). Impaired nuclear division caused by oryzalin likely doubled the DNA content and thus increased the nuclear size, as previously reported for the shoot apex (Grandjean et al, 2004). The PD-180970–treated seedlings also had enlarged nuclei, whereas the PP2-treated seedlings contained binuclear cells (Fig S2A). Moreover, compared with the regular cell layers in DMSO-treated seedlings, PD-180970–treated seedlings disrupted cellular organization, as indicated by distorted cell shape and large intercellular gaps that accumulated PI (Fig S2A). This is consistent with the general effects of PD-180970 on cortical MTs, rather than just the mitotic apparatus in Arabidopsis zygotes (Fig 1J). In contrast, PP2-treated seedlings retained the clear layered structure despite containing many binuclear cells. Combined with the proper alignment of MT and F-actin in PP2-treated zygotes, we concluded that PP2 is a specific inhibitor of cytokinesis.

We then assessed the long-term effects of these inhibitors by recording the root length daily in a 4-d treatment (Fig S2B). Despite their different effects on cells, treatment with PD-180970, PP2, and oryzalin similarly arrested root growth. To test the toxicity of the inhibitors, we compared the root length of seedlings grown in the presence or absence of the inhibitors after a 2-d inhibitor treatment (Fig S2C). The ratio of root lengths before and after the 5-d incubation was used as a growth indicator. Oryzalin-treated seedlings were unable to grow after inhibitor removal, probably because of the consumption of meristematic cells during treatment. In contrast, the PD-180970– and PP2-treated seedlings successfully resumed their growth (Fig S2C), indicating that these inhibitors did not cause irreversible damage to the cell viability.

**Protein identification using BY-2 cells for PD-180970 and PP2**

To investigate the inhibitory mechanisms of the compounds, we used the tobacco BY-2 cell system for subsequent proteomic analyses to identify the target proteins of PD-180970 and PP2. To obtain non-effective compounds as negative controls for proteomic analyses, we examined PD-180970 and PP2 analogs, which do not inhibit the division of BY-2 cells (Fig S3 and Table S1). Among the PD-180970 analogs, PD-166326 completely blocked cell division (100%, n = 27), whereas the PD-173955-Analog1 showed only faint effects (13%, n = 63), which were not significant compared with DMSO-treated samples (3%, n = 60, P = 0.1) (Fig S3A and Table S1). The same effects were observed in Arabidopsis (Fig S3C and Table S1). PD-166326 caused severe cell division arrest at the zygote, similar to PD-180970 (100%, n = 56), and PD-173955-Analog1 showed no effect (0%, n = 69). Therefore, we concluded that PD-166326 is as

a potent cell division inhibitor as PD-180970 and chose PD-173955-Analog1 as a negative control for PD-180970.

Among the PP2 analogs, only PP1 inhibited cell division (Fig S3B and C and Table S1). PP1 completely blocked cell division in BY-2 cells (100%, n = 32) and caused cell division arrest of Arabidopsis zygotes, although less frequently than PP2 (73%, n = 30). The other analogs had no inhibitory effect on cell division, as represented by PP3 (0%, n = 20 [BY-2 cells] and 0%, n = 46 [Arabidopsis embryos]; Fig S3B and C and Table S1). Therefore, we concluded that PP1 is also a potent cell division inhibitor and selected PP3 as the negative control for PP2. We also tested known inhibitors of Bcr-Abl (ponatinib, bosutinib, and bafetinib) and Src kinase (Src inhibitor1) in BY-2 cells and Arabidopsis embryos, and all exhibited zero or partial inhibition (0–70%; Fig S3A–C and Table S1). These results support our hypothesis that the inhibitory targets of PD-180970 and PP2 are plant-specific and independent of the Bcr-Abl and Src kinases found in animals.

PD-180970 and PP2 directly bind to core kinase pockets in animals and act as strong ATP-competitive inhibitors (Hanke et al, 1996; Zhu et al, 1999; Dorsey et al, 2000; La Rosee et al, 2002; Vajpai et al, 2008). Therefore, we hypothesized that these compounds target certain plant kinases and prevent the phosphorylation of particular substrates that are crucial for MT organization and phragmoplast formation (Fig 2A). These substrates are expected to be abundant in dividing cells, and their phosphorylation levels are reduced in the presence of PD-180970 and PP2. To identify substrates, we performed phosphoproteomics using BY-2 cells (Fig 2B). The BY–GTRC strain was synchronized at the mitosis (M) phase in the presence of the effective compound (PD-180970 or PP2) or ineffective controls (PD-173955-Analog1 or PP3), and whole cellular proteins were extracted. The phosphopeptides were then purified, and their sequences were determined using a high-sensitivity nanoLC-MS/MS system. We also generated a BY-2 protein reference database consisting of 50,171 sequences by converting the published transcriptome (RNA-seq) data obtained from non-transgenic BY-2 cells into amino acid sequences and determined the proteins found by phosphoproteomics (Fig 2B and Supplemental Data 1, Supplemental Data 2 and Supplemental Data 3) (Kozgunova et al, 2016). There were 14 and 12 proteins identified as the candidates for substrates of target kinases of PD-180970 and PP2, respectively, based on the criteria that the identified phosphopeptide number was 10 or more in the control cells with ineffective analogs and reduced to less than half in the presence of effective compounds (Tables S2 and S3, top five candidates are shown in Fig 2C and D, respectively). Candidate proteins were subjected to subsequent analyses to predict the target events of PD-180970 and PP2.

**Potential downstream targets for PD-180970**

Among the identified candidates for substrates of PD-180970 targets, RIBONUCLEASE PH45A-like (Nt-RRP45A-like) and MICROTUBULE-ASSOCIATED PROTEIN70-2-like (Nt-MAP70-2-like) showed no phosphopeptides in the presence of PD-180970, suggesting a strong inhibition (Fig 2C and Table S2). The Nt-RRP45A-like candidate exhibited similarity to the RRP45a, CER7, RRP42, and AT1G60080 Arabidopsis proteins, which are predicted to function in RNA-processing/

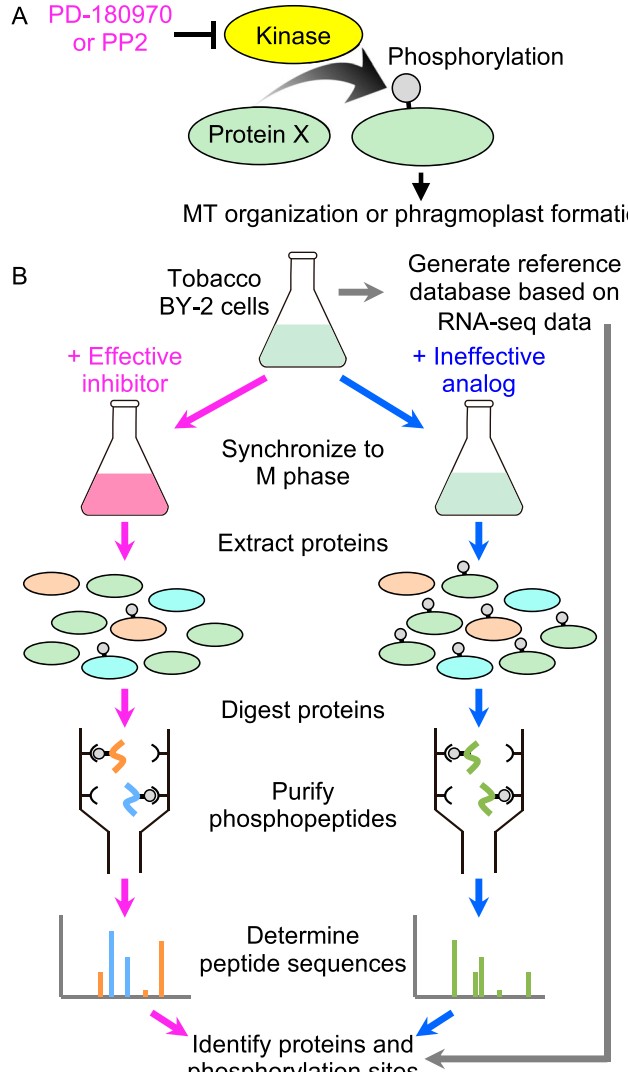

**C** Top candidates of PD-180970 phosphorylation target.

| | Peptide number | | | |
| | 1st | | 2nd | |
| Identified proteins in *N. tabacum* | - | + | - | + |
| --- | --- | --- | --- | --- |
| exosome complex component RRP45A-like | 15 | 0 | 11 | 0 |
| microtubule-associated protein (MAP) 70-2-like | 14 | 0 | 10 | 0 |
| pre-mRNA-processing protein 40A-like | 12 | 1 | 13 | 1 |
| proline-rich receptor-like protein kinase PERK8 | 10 | 2 | 10 | 3 |
| protein GLC8-like | 10 | 2 | 10 | 2 |

**D** Top candidates of PP2 phosphorylation target.

| | Peptide number | | | | | |
| | 1st | | 2nd | | 3rd | |
| Identified proteins in *N. tabacum* | - | + | - | + | - | + |
| --- | --- | --- | --- | --- | --- | --- |
| kinesin-like protein (Nt-KIN12A) | 192 | 112 | 195 | 43 | 205 | 50 |
| coatomer subunit alpha-1-like | 40 | 9 | 14 | 0 | 22 | 3 |
| MLO-like protein 1 | 45 | 6 | 13 | 0 | 20 | 0 |
| filament-like 6 isoform X3 | 69 | 29 | 25 | 8 | 58 | 23 |
| ethylene-insensitive protein 2-like | 39 | 12 | 22 | 3 | 41 | 14 |

**Figure 2. Target identification of PD-180970 and PP2.**
**(A)** Hypothetical model of the inhibitory mechanism of PD-180970 and PP2. **(A, B)** Schematic procedure of phosphoproteomics to identify the phosphorylation substrate ("protein X" in (A)). **(C, D)** Top five candidates of phosphorylation target of PD-180970 (C) and PP2 (D). **(C, D)** Peptide number shows the count of identified phosphopeptides in two experiments with effective PD-180970 (+) and ineffective PD-173955-Analog1 (–) (C), and in three experiments with effective PP2 (+) and ineffective PP3 (–) (D). All identified candidates and detailed data are shown in Tables S2 and S3 and Supplemental Data 2 and Supplemental Data 3.

degrading exosomes (Hooker et al, 2007). However, the identified Nt-RRP45A-like phosphorylation sites were not present in Arabidopsis proteins (Fig S4), which conflicts with the inhibitory effects observed in tobacco BY-2 and Arabidopsis.

In contrast, the three identified phosphorylated serine residues of Nt-MAP70-2-like were present in most Arabidopsis homologs (At-MAP70-1 to At-MAP70-5) (Fig S5A). Although the molecular functions of At-MAP70-2 to At-MAP70-4 remain unclear, At-MAP70-1 and At-MAP70-5 have reportedly decorated all MT structures, and it was revealed that At-MAP70-5 mediates the reorganization of MTs during lateral root formation (Korolev et al, 2005, 2007; Stöckle et al, 2022). Two of the three conserved phosphorylation sites were located in the essential region for MT association (Fig S5A) (Korolev et al, 2005). Therefore, we tested the effect of PD-180970 on MT structures using a high-resolution imaging system in BY-2 cells

(Murata et al, 2013). After 30–60 min of treatment with PD-180970 on MT/histone markers, the cortical MT, PPB, spindle, and phragmoplasts were severely disrupted (Fig 3A). This general effect supported our hypothesis that PD180970 disrupts MT organization as the primary effect via direct inhibition of MAP70 phosphorylation within the MT-associating motif. An examination of T-DNA insertion mutants of the five Arabidopsis *MAP70* genes showed no detectable defects in root growth (Fig S5B), implying a high gene redundancy or the presence of additional PD-180970 targets.

To investigate whether MAP70 protein colocalizes with MT and whether PD-180970 disrupts the localization, we generated a fluorescent marker of Nt-MAP70-2-like (MAP70 [Ser]) and transiently expressed it in BY-2 cells harboring MT/histone marker (Fig S6A and B). We found that MAP70 signals entirely merged with MT signals, but MT pattern itself was disrupted, suggesting the overdose effect

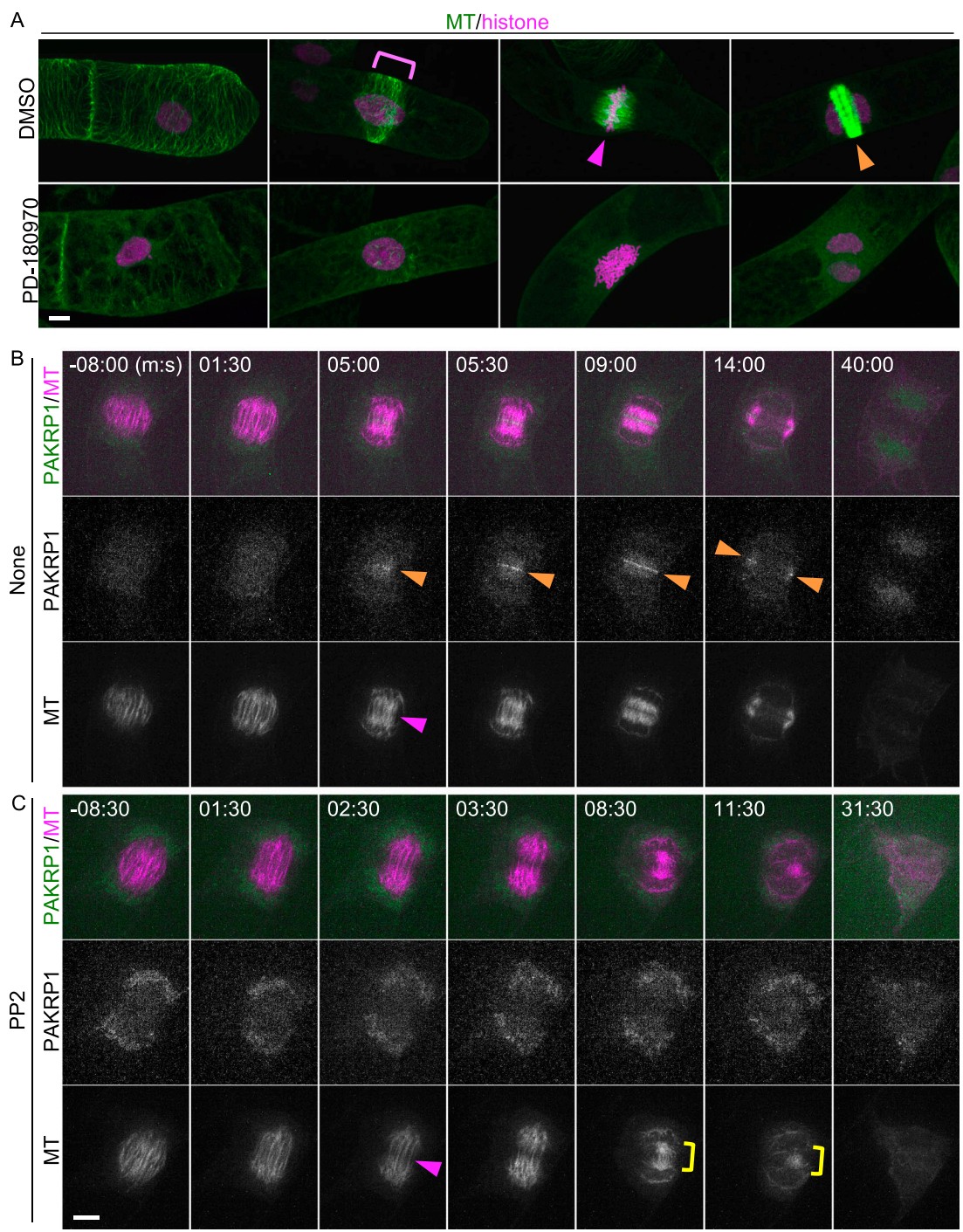

**Figure 3. PD-180970 disrupts MT organization, and PP2 blocks phragmoplast formation.**
**(A)** Confocal images of the BY-2 cells expressing MT/histone marker at 30–60 min after the application of indicated compounds. The magenta rectangle shows PPB. Magenta and orange arrowheads indicate the aligned chromatin at the center region of spindle, and the phragmoplast, respectively. **(B, C)** Time-lapse observation of the BY-2 cells expressing PAKRP1/MT marker in the presence of no compounds (B) and PP2 (C). Numbers indicate the time (min:sec) from the anaphase onset. The upper, middle, and lower panels show the merged, PAKRP1, and MT images, respectively. Magenta and orange arrowheads indicate the central gap region of spindle and the PAKRP1 localization on phragmoplast, respectively. Yellow rectangle shows the remnant MT bundles connecting two sister chromatids. Scale bars: 10 μm.

of MAP70 on MT organization (Fig S6B). We also exchanged all three serine residues that we identified with alanines to block phosphorylation (MAP70 [Ala]) and with aspartates to mimic constitutively active phosphorylation (MAP70 [Asp]), and both showed the same result to MAP70 (Ser) (Fig S6C and D). These results showed that highly expressed MAP70 could bind to MT and interfere with its organization regardless of the phosphorylation state of these three serines.

None of the other candidates showed a direct relationship to MT organization, according to the known functions of themselves and similar proteins in Arabidopsis (Table S2). However, they might have an indirect contribution, such as the protein degradation of key regulators via 26S proteasome and ubiquitin (candidates 13 and 14) (Kurepa & Smalle, 2008), and mRNA processing via PRE-MRNA-PROCESSING PROTEIN 40A (PRP40A), PRP40B, and PRP40C (candidate 3) (Kang et al, 2009). It is also possible that alterations in plasma membrane properties via PATL1/2-dependent membrane trafficking (candidate 6) (Zhou et al, 2019) and the endocytosis adaptor protein complex TPLATE (candidates 7 and 9) (Gadeyne et al, 2014; Wang et al, 2020) may affect cortical MT patterns.

## Potential downstream targets for PP2

Among the identified candidates for substrates of PP2 targets, KINESIN-12A (Nt-KIN12A) showed over 100 phosphopeptides in the control cells (Fig 2D and Table S3), with protein abundance and/or high phosphorylation levels in the dividing cells. In Arabidopsis, PHRAGMOPLAST-ASSOCIATED KINESIN-RELATED PROTEIN1 (PAKRP1)/At-KIN12A and PAKRP1L/At-KIN12B were the proteins most similar to Nt-KIN12A (Fig S7). PAKRP1 and PAKRP1L belong to the class II Kinesin-12 family with At-KIN12F, whose function remains unidentified (Müller & Livanos, 2019). Kinesin-12 members have MT plus end-directed motility, and PAKRP1 and PAKRP1L are essential for cell plate formation, as *pakrp1 pakrp1l* double mutant disturbs the first postmeiotic cytokinesis owing to disorganization of phragmoplast MTs (Lee et al, 2007). Twelve phosphorylated residues were identified in Nt-KIN12A, and six were conserved in PAKRP1 or PAKRP1L (Fig S7). Five conserved residues were found between the putative kinesin motor domain and coiled-coil region (Fig S7), implying a regulatory element for protein activity and/or interaction (Ems-McClung & Walczak, 2010; Mann & Wadsworth, 2019).

To test whether PP2 disrupts the localization of PAKRP1 and PAKRP1L on phragmoplasts, we generated fluorescent markers and observed colocalization with MT in the presence and absence of the inhibitor using a high-speed time-lapse system of BY-2 cells (PAKRP1/MT, Fig 3B and Video 4, and PAKRP1L/MT, Video 5) (Murata et al, 2013; Murata & Baskin, 2014). In the absence of PP2, both proteins appeared at the central gap region in the remnant spindle, where phragmoplasts emerged (Fig 3B, Video 4, and Video 5). Both proteins associated with the expanding phragmoplasts then disappeared upon completion. In the presence of PP2, the central gap region was detected, but PAKRP1 and PAKRP1L did not accumulate at this site (Fig 3C, Video 4, and Video 5). The proper phragmoplasts were not formed throughout the entire process, and the MT bundles were abnormally concentrated in the cell center. These results show that PAKRP1 and PAKRP1L localization was abolished by PP2 as early as the phragmoplast initiation phase. We propose that PP2 blocks phragmoplast formation primarily by inhibiting the phosphorylation of class II Kinesin-12.

We then exchanged all nine serine residues, wherein we found phosphorylation in PAKRP1, with alanines [PAKRP1 (Ala)] and with aspartates (PAKRP1 [Asp]), and transiently expressed them in BY-2 cells harboring MT/histone marker (Fig S8, Video 6, and Video 7). As similar to the result of PAKRP1/MT (Fig 3B and C), the unmodified PAKRP1 (PAKRP1 [Ser]) localized on expanding phragmoplast and

PP2 blocked this localization and phragmoplast formation (Fig S8A and Video 6). PAKRP1 (Asp) exhibited similar localization and was also abolished by PP2 treatment (Fig S8B and Video 7). This result suggests either that PAKRP1 (Asp) could not successfully mimic the constitutive phosphorylation, or that there are additional phosphorylation sites on PAKRP1 or other PP2 downstream target proteins are involved. On the contrary, PAKRP1 (Ala) signals merged with the entire cortical MT, not specifically with phragmoplast, and MT pattern was disrupted as found in overexpressed MAP70 markers (Fig S8C, compare with Fig S6). This suggests that highly expressed phospho-blocked PAKRP1 proteins were too much stabilized on MT and thus disrupted MT organization.

Among other candidates, the coatomer subunit α-1-like of Coat Protein I (COPI) complex (candidate 2) may also contribute to cytokinesis (Fig 2D and Table S3). The COPI complex is responsible for retrograde transport from Golgi to ER and intra-Golgi transport (Paul & Frigerio, 2007). In tobacco BY-2 cells, it was reported that depletion of the coatomer subunit disrupts the Golgi structure and accumulation of autolysosome-like structures, as well as failure of cell plate formation (Hee-Kyung et al, 2015). However, because phragmoplasts were formed normally in this situation, it was presumed that the COPI complex contributes to cytokinesis via vesicular transport that supplies cell plate components (Hee-Kyung et al, 2015). These defects are not consistent with the specific effect of PP2 on phragmoplast formation.

The filament-like 6 (candidate 4) exhibited similarity to TRICHOME CELL SHAPE1 (TCS1) in Arabidopsis proteins, which is required for cortical MT stability and thus for proper trichome branching (Chen et al, 2016). Therefore, it might be involved in MT organization in phragmoplast, although the role of TCS1 in cytokinesis has not been reported. Other candidates could also play indirect roles, such as turnover of regulatory proteins via ubiquitin-dependent degradation (candidate 11) (Li et al, 2022) or altered intracellular trafficking via MILDEW RESISTANCE LOCUS O (MLO) proteins, which show dynamic redistribution in endomembrane system upon signals (candidate 3) (Bhat et al, 2005).

## PD-180970 and PP2 are effective in diverse plant species

To test whether PD-180970 and PP2 could be used in other plant species, we analyzed root length after inhibitor treatments in tobacco (*Nicotiana benthamiana*) and cucumber (*Cucumis sativus*). As observed for Arabidopsis (Fig S2B), all compounds strongly interfered with root growth in both species (Fig S9A and B). In addition to these angiosperm species, we also tested the moss *Physcomitrium patens*, which diverged from angiosperms at least 500 million yr ago (Morris et al, 2018). Similar to oryzalin, PD-180970 effectively perturbed MT organization, as the filamentous cortical arrays of MT markers were disrupted in the chloronema tip cells of *P. patens* (Fig 4A) (Kozgunova & Goshima, 2019). Free GFP–tubulin signals were detected in the cytoplasm, and chloroplast shapes were highlighted in compound-treated cells. This result suggests that PD-180970 affects MT in moss in a manner similar to that observed in Arabidopsis zygotes and tobacco BY-2 cells (Figs 1H–J and 3A, respectively).

To examine whether PP2 affects class II Kinesin-12 protein in the moss, we observed Pp-Kinesin-12IIc, which localizes at the

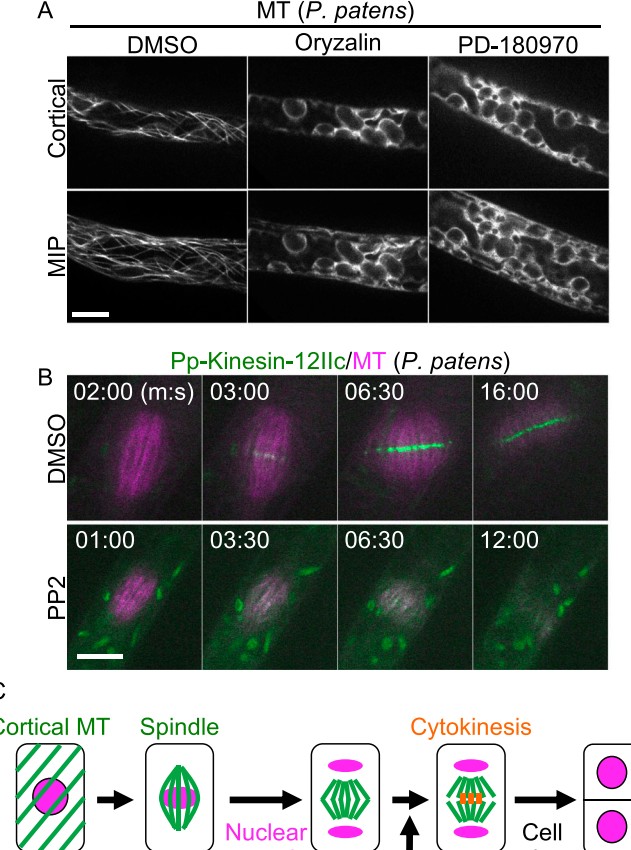

**Figure 4. Both PD-180970 and PP2 are effective in *P. patens*.**
**(A)** Confocal images of *P. patens* chloronema tip cells expressing MT marker at 30 min after the application of indicated compounds. Upper panels show the images of the cortical region, and lower panels display MIP images. **(B)** Time-lapse observation of *P. patens* chloronema tip cells expressing Pp-Kinesin-12IIc/MT marker in the presence of indicated compounds. Numbers indicate the time (min:sec) from the anaphase onset. **(C)** Schematic representation of the inhibitory dynamics of PD-180970 and PP2 during plant cell division. Scale bars: 10 μm.

phragmoplast midzone (Pp-Kinesin-12Iic/MT; Fig 4B and Video 8) (Miki et al, 2014). Pp-Kinesin-12Iic localization was altered by PP2 application, as the strong accumulation at the phragmoplast midzone disappeared (Fig 4B and Video 8), consistent with tobacco BY-2 cells (Fig 3B and C). In the presence of PP2, proper phragmoplast formation was abolished and no cell plate was formed, resulting in binuclear cells and subsequent nuclear fusion (Fig S9C and Video 9); both of these results were observed in the BY-2 and Arabidopsis zygotes (Figs 1G and S1B). We conclude that PD-180970 and PP2 effectively block MT organization and phragmoplast formation, respectively, and were applicable to diverse plant species (Fig 4C).

# Discussion

In this study, we identified cell division inhibitors by combining a chemical screening in Arabidopsis zygotes, specification of target events using tobacco BY-2 cells, and validation of their effectiveness in multiple plant species. According to our high-resolution time-lapse imaging and phosphoproteomics, two identified inhibitors, PD-180970 and PP2, were specific to the MT alignment and phragmoplast formation, respectively. In addition, their inhibitory effects were non-lethal and effective in various cell types and plant species. These properties make PD-180970 and PP2 useful tools for future cell division studies, as they provide novel manipulation nodes.

In agreement with the reports that PD-180970 and PP2 act as kinase inhibitors in animals, we found that PD-180970 and PP2 reduced the phosphorylation levels of diverse proteins. In particular, we identified MAP70 proteins, representing a potential downstream target of PD-180970. MAP70 is an MT-associated factor (Morejohn et al, 1987), but its molecular functions are still unknown, except for Arabidopsis MAP70-5. At-MAP70-5 was reported to increase MT length in vitro and to reorganize cortical MTs to alter the endodermal cell shape for lateral root initiation, suggesting that At-MAP70-5 mediates dynamic change of MT arrays (Korolev et al, 2007; Stöckle et al, 2022). Our phosphoproteomic analysis identified several phosphorylated sites in the conserved MT-binding domain of the MAP70 family (Korolev et al, 2005). These findings suggest that PD-180970 would block yet-unidentified kinases that phosphorylate mainly MAP70 to disrupt MT organization (Fig 4C), similar to Aurora kinase, which phosphorylates MAP65-1 to activate its MT-bundling capacity (Boruc et al, 2017). Yet, we cannot exclude the possibility that PD-180970 disrupts MT organization by stabilizing MAP70 (and other proteins) on MT without affecting phosphorylation because MAP70 overexpression caused MT disorganization regardless of the phosphorylation state and PD-180970 rapidly disrupted MTs only 30 min after application (Figs 3, 4, and S7).

It would be difficult to visualize the PD-180970's effect on the MT-binding ability of MAP70 because MT alignment itself was destroyed by both the application of PD-180970 and the expression of MAP70 markers (Figs 3 and S7). More detailed investigations are necessary to reveal the relationship between PD-180970, MAP70, and MT organization. For example, it would be useful to identify the direct binding targets of PD-180970. Such analysis will also help to detail the molecular function of MAP70 proteins, which are presumably important proteins localized on entire MT structures (Korolev et al, 2005, 2007) and are difficult to assess using a genetic approach, owing to the lack of a detectable mutant phenotype (Fig S5). In addition, we cannot exclude the additional contributions of other candidate proteins whose phosphorylation levels were reduced in the presence of PD-180970 (Fig 2 and Table S2). However, we could not find any direct relationship between these proteins and MT organization in the literature.

We predicted class II Kinesin-12 as the top candidate for PP2 through phosphorylation inhibition. In tobacco BY-2 cells and *P. patens*, PP2 blocked the accumulation of class II Kinesin-12 proteins in the central region of the remnant spindle from the beginning of phragmoplast initiation (Figs 3 and 4). In Arabidopsis, a double

mutant lacking PAKRP1 and PAKRP1L did not form phragmoplast MT during the first postmeiotic cytokinesis (Lee et al, 2007). These observations followed the strong inhibition of PP2 on cytokinesis, thereby suggesting class II Kinesin-12 is a potential downstream target of PP2 (Fig 4C). Yet, the fact that the substitution of the nine serine residues for aspartic acid could not prevent the phragmoplast destruction by PP2 suggests that additional phosphorylation sites or other candidate downstream targets may be present (Fig S8). In addition to class II, the Kinesin-12 family contains three members of class I: PHRAGMOPLAST ORIENTING KINESINS (POK1)/At-KIN12C, POK2/At-KIN12D, and At-KIN12E (Müller & Livanos, 2019). POK1/POK2 and At-KIN12E are required for cell plate orientation and spindle assembly, respectively, but they also may localize to the phragmoplast midzone (Herrmann et al, 2018, 2021; Müller & Livanos, 2019). To understand the inhibitory mechanism and specificity of PP2, it would be useful to examine whether PP2 also affects class I members and another class II member, At-KIN12F, and to identify the direct binding targets of PP2.

In Arabidopsis, PAKRP1 and PAKRP1L interact with the TWO-IN-ONE (TIO)/FUSED kinase (Oh et al, 2012). Although it remains unclear whether TIO phosphorylates these proteins, the *tio* mutant and *pakrp1 pakrp1l* double mutant failed to form a cell plate during male gametogenesis, suggesting their cooperative function in cytokinesis (Oh et al, 2005, 2014; Lee et al, 2007). Therefore, it is important to assess whether TIO is the PP2's direct binding target to mediate class II Kinesin-12 phosphorylation during phragmoplast formation.

In contrast to the gametogenetic defects of the *pakrp1 pakrp1l* double mutant, individual mutants showed no detectable defects (Pan et al, 2004; Lee et al, 2007). This gene redundancy and mutant lethality of key regulators prohibit our genetic approaches from analyzing the molecular mechanisms of specific cell division events. As a result, even the core regulations, such as how phragmoplasts are initiated and how membrane trafficking toward the cell plate is promoted, are still poorly understood. PD-180970 and PP2 showed strong and reversible effects, suggesting that they temporally block related targets simultaneously, thereby circumventing redundancy, lethality, and secondary effects. Therefore, we believe these compounds are powerful tools for investigating the detailed mechanisms of MT organization and phragmoplast formation.

## Materials and Methods

### Growth conditions and plant strains

Arabidopsis and *N. benthamiana* were grown in petri dishes containing a 1.5% agar medium and 1/2 Murashige and Skoog (MS) medium or on the soil at 18–22°C under continuous light or long-day conditions (16-h light/8-h dark). Tobacco BY-2 cells and *P. patens* strains were cultured as described previously (Kurihara et al, 2008; Yamada et al, 2016). The "Chinese long" strain was used as *C. sativus* and grown in petri dishes containing 0.1% PLANT PRESERVATIVE MIXTURE (PPM; Nacalai Tesque) as antibacterial reagent under continuous light at 18–22°C.

All Arabidopsis lines were placed in a Columbia (Col-0) background. Mutants of *at-map70-1* (SALK_013866), *at-map70-2* (SALK_060997), *at-map70-3* (SALK_203128), *at-map70-4* (SALK_069552), and *at-map70-5* (SALK_106968) were identified in the SALK institute.

### Compounds

The following compounds were used: oryzalin (36182; Sigma-Aldrich), 5-iodotubercidin (5-ITu) (I100; Sigma-Aldrich), PD-180970 (PZ0142; Sigma-Aldrich), PD-166326 (9000988; Cayman Chemical), PD-173955-Analog1 (SYN-1062; SYNkinase), ponatinib (CS-0204; CHEMSCENE), bosutinib (PZ0192; Sigma-Aldrich), bafetinib (A10119; AdooQ Bioscience), PP2 (P0042; Sigma-Aldrich), PP1 (BML-EI275; Enzo Life Sciences), 4-aminopyrazolo[3,4-d]pyrimidine (PP2-Analog1) (A1041; Tokyo Chemical Industry), 1-tert-butyl-1H-pyrazolo[3,4-d]pyrimidine-4-amine (PP2-Analog2) (GF-0723; Key Organics), PP3 (A2737; Tokyo Chemical Industry), and Src inhibitor1 (sc-204303; Santa Cruz Biotechnology) (Karni et al, 2002; Wisniewski et al, 2002; Hum et al, 2014; Rossari et al, 2018).

Each compound was dissolved in DMSO and used at a final concentration of 10 $\mu$M in 0.1% DMSO, except for the dosage assay, and in oryzalin, which was used at 1 $\mu$M. Oryzalin (10 $\mu$M) was used in the *P. patens* assay.

### Fluorescent markers and microscopy for Arabidopsis

In the histone/PM marker for Arabidopsis zygotes and embryos, green fluorescent protein fused to histone H2B and red fluorescent protein fused to LOW-TEMPERATURE-INDUCED6b were expressed under a zygote/embryo-specific promoter of *WUSCHEL RELATED HOMEOBOX2* gene (WOX2p::H2B–GFP and WOX2p::tdTomato–LTI6b) (Gooh et al, 2015). Live-cell imaging of this histone/PM marker was performed using an inverted confocal microscope system (CV1000; Yokogawa Electric) as described previously (Nambo et al, 2016).

MT and F-actin were observed in Arabidopsis zygotes using EC1p::Clover–TUA6 (Kimata et al, 2016) and EC1p::Lifeact–Venus (Kawashima et al, 2014). Live-cell imaging of these markers was performed using a two-photon excitation microscope (LSM780-DUO-NLO; Zeiss) as previously described (Kurihara et al, 2017; Ueda et al, 2020).

For the observation of root meristem in Arabidopsis, the histone marker RPS5Ap::H2B–GFP (Maruyama et al, 2013) was stained with 10 $\mu$g/ml PI (Sigma-Aldrich) to visualize the plasma membrane. The samples were observed using an LSM780-DUO-NLO (Zeiss) as described previously (Kimata et al, 2019).

### Fluorescent markers and microscopy for tobacco BY-2 cells

As the MT/centromere marker for tobacco BY-2 cell culture cells, we used the BY–GTRC strain, which labeled MT and centromeric histone by expressing GFP-fused $\alpha$-tubulin and RFP (red fluorescent protein)-labeled CenH3 (centromeric histone H3) (35S::GFP–$\alpha$-tubulin and 35S::RFP–CenH3) (Kurihara et al, 2008). This marker was observed using a confocal microscope system (CV1000; Yokogawa Electric) as described previously (Nambo et al, 2016).

As MT/histone markers for high-resolution imaging of BY-2 cells, we used NOSp::mCitrine–TUB8 and H2Bp::H2B–mCherry. These markers were tandemly cloned into the pCAMBIA1300 binary vector via *P. patens*

intergenic (PIG) 1L region (Okano et al, 2009). For NOSp::mCitrine–TUB8, a 0.18-kb NOS promoter was fused to yellow fluorescent protein mCitrine, NtTUB8 (LOC107786877) cDNA, and a 0.25-kb AtHSP18.2 (AT5G59720) terminator. For H2Bp::H2B–mCherry, red fluorescent protein mCherry was inserted before the stop codon of genomic sequence of *N. tabacum* histone H2B spanning from 1.9-kb 5′-UTR to 0.5-kb 3′-UTR. Images were acquired as z-stacks with 0.8-$\mu$m intervals using a confocal microscope system (FV3000; Olympus) equipped with two GaAsP detectors and a water immersion 60× objective lens (NA 1.2). The emission signal of mCitrine was detected between 500 and 550 nm with a 488-nm excitation, and that of mCherry was detected between 600 and 680 nm with a 561-nm excitation. Control DMSO or PD-180970 was applied between 30 and 60 min before observation.

To observe the MT array and Kinesin-12 localization in BY-2 cells, NOSp::mScarlet-i–TUB8 was combined with PAKRP1 and PAKRP1L markers (PAKRP1/MT and PAKRP1L/MT). In NOSp::mScarlet-i–TUB8, the NOS promoter was fused to the red fluorescent protein mScarlet-i, NtTUB8 cDNA, and the 0.47-kb NtEF-1-$\alpha$ terminator in a pRI910 binary vector. The PAKRP1 and PAKRP1L markers used were RPS5Ap::Clover–PAKRP1 (MU2384) and RPS5Ap::Clover–PAKRP1L (MU2403), respectively. Among these markers, a 1.7-kb RIBOSOMAL PROTEIN SUBUNIT 5A (RPS5A) promoter (Adachi et al, 2011) was fused to the green fluorescent protein Clover, the full-length coding region of PAKRP1 (AT4G14150) or PAKRP1L (AT3G23670), and the NOPALINE SYNTHASE (NOS) terminator in a pMDC99 binary vector (Curtis & Grossniklaus, 2003). Images were acquired as previously described using an inverted microscope (IX81; Olympus) equipped with a spinning-disk unit (CSU21; Yokogawa) and a water immersion 60× objective lens (NA 1.2), with an additional optical unit (W-view Gemini; Hamamatsu) used for simultaneous image acquisition of green and red channels (Murata et al, 2013). The high-speed time-lapse images of the PAKRP1/MT and PAKRP1L/MT markers were acquired every 30 s with band-pass filters (Semrock FF01-520/60-25 and FF01-609/54-25 for mClover and mScarlet-i, respectively), and the images were expanded by an additional 1.5× extension lens in front of the camera. The compounds were applied during live-cell imaging as described previously (Murata & Baskin, 2014).

As a background for the transient expression of MAP70 or PAKRP1 mutants in BY-2 cells, NOSp::mScarlet-i–TUB8 and H2Bp::H2B-mCherry were tandemly cloned into the pCAMBIA1300 binary vector via PIG1L region (MT/histone). To generate MAP70 or PAKRP1 mutants, the predicted phosphorylation sites were substituted with alanine or aspartate using artificial DNA synthesis by Twist Bioscience (https://www.twistbioscience.com/). To construct NOSp::mClover-MAP70 (Ser/Ala/Asp, coded as YK58-60) and NOSp::mClover-PAKRP1 (Ser/Ala/Asp, coded as MU2566, YK61, and YK62), synthesized DNA fragments were fused to mClover and inserted between NOS promoter and ribulose-1,5-bisphosphate carboxylase (Rubisco) small subunit (rbcS) terminator on a pRI910 binary vector.

For transient transformation, 4 ml 4-d-old culture was cocultivated with 100 $\mu$l *Agrobacterium tumefaciens* LBA4404 strain harboring each construct at 26°C in a culture dish. After 2 d, the cells were washed with 3% sucrose and selected on agar medium containing 50 mg/l kanamycin and 50 mg/l carbenicillin. After 5 d, surviving cells were resuspended in a liquid medium and subjected to microscopy. Images were acquired using a two-photon microscope (A1 MP; Nikon) equipped with Ti:sapphire femtosecond pulse laser

(Mai Tai DeepSee; Spectra-Physics) and a GaAsP detector. Fluorescent signals were detected using a water immersion 40× objective lens (CFI Apo LWD WI, NA = 1.15, WD = 0.59–0.61 mm; Nikon) and two band-pass filters (534/30 nm for Clover, and 578/105 nm for mScarlet-i and mCherry). mClover-MAP70 markers were observed as z-stacks with 1-$\mu$m z-interval at 930-nm excitation and 2× zoom, whereas mClover-PAKRP1 markers were observed as time series with 5-min interval of single z-plane at 950-nm excitation and 3× zoom. PP2 was applied just before the observation.

### Fluorescent markers and microscopy for *P. patens*

*P. patens* chloronema was observed using the MT marker (PpGCP4p:: GFP–tubulin) (Kozgunova & Goshima, 2019), the dual-color marker of MT and nucleus (PpGCP4p::GFP–tubulin and 7113p:: histone H2B–mRFP) (Kozgunova & Goshima, 2019), or Pp-Kinesin-12IIc and MT (Pp-Kinesin-12IIc–Citrine and PpACTp::mCherry–tubulin) (Miki et al, 2014).

For the PD-180970 experiment, chloronema tissues expressing MT marker cultured on a cellophane-laid BCDAT plate for 6 d were sonicated in a BCD liquid medium containing 10 $\mu$M oryzalin, 10 $\mu$M PD-180970, or 0.5% DMSO, followed by incubation for 30 min. The tissues were introduced into microfluidic devices and immediately observed (Kozgunova & Goshima, 2019). The images were acquired with an inverted microscope (Ti, 100 × 1.45 NA lens; Nikon) equipped with a spinning-disk confocal unit (CSU-X1; Yokogawa), 488- and 561-nm laser lines (LDSYS-488/561-50-YHQSP3, Pneum), and an electron-multiplying charge-coupled device camera (ImagEM; Hamamatsu) at 2.5-$\mu$m z-intervals. The microscope was controlled using NIS-Elements.

For PP2 experiments, chloronema tissues were cultured in six-well glass-bottom dishes or 35-mm dishes in a BCD agarose medium for 5–7 d (Yamada et al, 2016). Water containing 10 $\mu$M PP2 or 0.5% DMSO was directly applied to the dishes, and the mosses were incubated for 30 min before observation. High-resolution live-cell imaging of the Pp-Kinesin-12IIc/MT marker was performed using the same microscope described above. Long-term imaging of MT/ histone markers was performed with a wide-field microscope (TE2000, 10 × 0.45 NA lens; Nikon) equipped with a CMOS camera (ZYLA-4.2P-USB3; Andor) and a Nikon Intensilight Epi-fluorescence illuminator, which was controlled by iQ software.

### Chemical screening

Young ovules were collected from the siliques of ~5 mm and cultivated in 96-well glass-bottom plates with individual compounds at 10 $\mu$M from the LOPAC Pfizer library (LO5100; Sigma-Aldrich) and SCREEN-WELL Kinase Inhibitor library (BML-2832; Enzo Life Sciences). After incubation for 2 d in ovule cultivation media (Gooh et al, 2015; Kurihara et al, 2017), the ovules were observed using an inverted fluorescent microscope (IX73; Olympus).

### Phosphoproteomics using synchronized BY-2 culture cells

For phosphoproteomics, BY–GTRC cells at 7 d after transfer to fresh medium were synchronized at the DNA replication stage (synthesis [S] phase) as described previously (Nagata & Kumagai, 1999;

Kumagai-Sano et al, 2006). The cells were cultured in the presence of PD-180970, PD-173955-Analog1, PP2, or PP3 for 8–9 h. After confirming that most cells started mitosis using an upright microscope (Axi-oImager A2; Zeiss), total proteins were extracted using cell lysis buffer (50 mM Tris–HCl [pH 8.0], 150 mM NaCl, 1% [vol/vol] Triton X-100, 25 µM MG-132, and cOmplete Mini Protease Inhibitor Cocktail [Roche]). After the extraction of crude proteins and trypsin digestion, peptides were purified using an immobilized metal ion affinity chromatography column or a sequential enrichment of immobilized metal affinity chromatography column, both of which specifically absorb phosphopeptides. Their amino acid sequences were determined using a high-sensitivity nanoLC-MS/MS system, as previously described (Ohkubo et al, 2021).

The protein sequences of BY-2 cells were predicted based on the transcriptome (RNA-seq) data, which have been previously reported (Kozgunova et al, 2016). Each of the transcript data was converted to an amino acid sequence, and a total of 50,171 protein sequences (coded from NtBYT000000.000 to NtBYT078147.000 in Supplemental Data 1) were used as a reference database to map the identified phosphopeptides by MASCOT search to determine the corresponding proteins. Identified proteins and their homologous proteins were aligned using Clustal Omega software (https://www.ebi.ac.uk/Tools/msa/clustalo/). Predictions of the MT-binding region in MAP70 proteins have been previously reported (Korolev et al, 2005). Kinesin motor domains and coiled-coil domains in PAKRP1 and PAKRP1L were predicted using UniProt (https://www.uniprot.org).

## Data Availability

Source data used for the figures are presented in Supplemental Data 1, Supplemental Data 2 and Supplemental Data 3. All other data are available from the corresponding author upon request.

## Supplementary Information

## Acknowledgements

We thank Tomomi Yamada, Yumi Kuwabara, Terumi Nishii, and Azusa Utsumi for their technical support, and Michiko Sasabe, Masaki Ito, Gohta Goshima, Yoshikatsu Sato, Simon Miller, Tsuyoshi Hirota, Shunsuke Oishi, Yasunori Machida, and Ping Kao for helpful discussions. We thank Takeshi Kuroha, Mina Ohtsu, and Michitaka Notaguchi for providing seeds of various plant species and Toshinori Kinoshita and Koji Takahashi for providing the chemical library. Microscopy was supported by the Live Imaging Center at the Institute of Transformative Bio-Molecules (WPI-ITbM) of Nagoya University. This work was supported by Japan Advanced Plant Science Network, Japan Society for the Promotion of Science (Grant-in-Aid for Early-Career Scientists [JP21K15117 to Y Kimata], Grant-in-Aid for Research Activity Start-up [19K23723 to M Yamada], Grant-in-Aid for Scientific Research on Innovative Areas [19H04859, 19H05670, and 19H05676 to M Ueda; JP16H06465, JP16H06464, and JP16K21727 to T Higashiyama; and JP16H06280 [Advanced Bioimaging Support], JP20H05358, and JP22H04668 to D Kurihara], Grant-in-Aid for Scientific Research [B] [JP19H03243 to M Ueda and JP18H02471 to T Murata], Grant-in-Aid for Scientific Research [S] [JP16H06378 to T Murata and M Hasebe], Grant-in-Aid for Challenging Exploratory Research [JP19K22421 to M Ueda]), the Japan Science and Technology Agency (PRESTO Programme [JPMJPR18K4 to D Kurihara], FOREST Programme [JPMJFR204T to D Kurihara], and CREST [JPMJCR2121 to M Ueda]), Suntory Rising Stars Encouragement Program in Life Sciences (SunRiSE) to M Ueda, and Toray Science Foundation to M Ueda (20–6102).

## Author Contributions

Y Kimata: conceptualization, data curation, funding acquisition, investigation, visualization, and writing—original draft, review, and editing.
M Yamada: funding acquisition, investigation, visualization, and writing—original draft, review, and editing.
T Murata: funding acquisition, investigation, visualization, and writing—original draft.
K Kuwata: data curation and investigation.
A Sato: conceptualization.
T Suzuki: investigation.
D Kurihara: conceptualization, funding acquisition, investigation, and writing—original draft.
M Hasebe: funding acquisition and investigation.
T Higashiyama: conceptualization and funding acquisition.
M Ueda: conceptualization, data curation, supervision, funding acquisition, investigation, visualization, project administration, and writing—original draft, review, and editing.

### Conflict of Interest Statement

The authors declare that they have no conflict of interest.

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
