## [Reviewer comments · Life Science Alliance]

Life Science Alliance

Novel inhibitors of microtubule organization and phragmoplast formation in diverse plant species

Yusuke Kimata, Moé Yamada, Takashi Murata, Keiko Kuwata, Ayato Sato, Takamasa Suzuki, Daisuke Kurihara, Mitsuyasu Hasebe, Tetsuya Higashiyama, and Minako Ueda

DOI: <https://doi.org/10.26508/lsa.202201657>

Corresponding author(s): Minako Ueda, Tohoku University

Review Timeline:

Submission Date:	2022-08-08
Editorial Decision:	2022-08-08
Revision Received:	2023-01-19
Editorial Decision:	2023-02-13
Revision Received:	2023-02-14
Accepted:	2023-02-14

Transaction Report:

Please note that the manuscript was reviewed at Review Commons and these reports were taken into account in the decision-making process at Life Science Alliance.

Manuscript number: RC-2022-01469

Corresponding author(s): Minako, Ueda

1. General Statements

We really appreciate the positive comments and suggestions of the reviewers on our submitted manuscript. We think we will be able to solve the issues inquired by reviewers by adding new data and revising the phrases as detailed below.

2. Description of the planned revisions

Reviewer #1:

Major comments

Localization analysis of a transiently expressed MAP70 transgene with inactivating phosphosite mutations would be important to see whether the identified conserved phosphosites are relevant for MAP70 interaction with MTs. This experiment could be performed rapidly using transient expression in BY-2 cells.

We agree on the importance of this analysis. Therefore we are currently preparing fluorescent markers of Nt-MAP70-2-like and its phospho-blocked (Ala) version to coexpress with MT and nuclear markers in BY-2 cells. We estimate that we need three more months to complete this experiment.

The authors propose that PP2 blocks phragmoplast formation by preventing phosphorylation of class II Kinesin-12 proteins. In support, authors show that PP2 treatment correlates with a decrease in KIN12A phosphopeptide count (not fully abolished) and its failure to localize to emerging phragmoplasts in BY-2 cells and Physcomitrium. As class II Kinesin-12 proteins have been previously implicated in phragmoplast assembly this is a fairly reasonable hypothesis, but would benefit from the analysis of transgenic KIN12A variants carrying inactivating (A) or potentially activating (D/E) phosphosite mutations. Is loss of phosphorylation sufficient to prevent phragmoplast localization? Can an activated variant rescue PP2-induced KIN12A localization and cell division defects? As above, using transient expression in BY-2 cells would be a fast approach to tackle these questions.

We are currently preparing fluorescent markers of phospho-blocked (Ala) and phospho-mimic (Asp) versions of KIN12A (PAKRP1) to coexpress with MT and nuclear markers in BY-2 cells. We will check whether they localize to phragmoplast and also test PP2 effects. We would need three more months to complete these analyses.

Reviewer #2:

Revision Plan

Major comments

- The manuscript would strongly benefit from being revised by a native English speaker. There are many unusual or awkward formulations, in particular in the abstract.

We apologize for unnatural sentences. After adding new data and correcting the manuscript, we will ask a native English speaker to revise it.

Reviewer #3:

Major comments

The major concern is lack of evidence to connect MAP70 and MT disruption upon treatment with PD-180970, in contrast to PP2, which was shown to affect localization of Kinesin-12. I wonder if authors could use taxol to stabilize MTs, then observe the localization of MAP70 with application of PD-180970?

As we responded to reviewer 1, we are preparing the fluorescent marker of Nt-MAP70-2-like to coexpress with MT and nuclear markers in BY-2 cells. By using this multi-color marker, we will test whether PD-180970 affects the localization of MAP70 on MTs, also using taxol. However, in our experience, taxol is not a very effective inhibitor and may not work in our transient expression system in BY-2 cells. In that case, we will analyze whether phospho-mimic (Asp) version can prevent MT disruption in the presence of PD-180970 to assess the relation of PD-180970, MAP70 and MT disruption.

I have another concern on the action of PD-180970. PD-180970 appears to affect ubiquitously indispensable proteins for MTs. If PD-180970 disrupts MT by inhibiting phosphorylation of some MAPs, it must need time for turnover of proteins phosphorylated before PD-180970 was applied. In the proteomics experiment, authors treated the cells with the compounds for 8-9 hr. On the other hand, in BY-2 cells, PD-180970 disrupted MTs only 30 min after application of PD-180970. I wonder if proteins were replaced during the 30 min. Could authors examine how long it takes to affect interphase MTs? If PD-180970 disrupts MTs in a 5-10 min like oryzalin, it is unlikely that inhibition of phosphorylation of proteins like MAP70 caused MT disruption. Rather, it may inhibit some proteins that have activity to disrupt microtubules but are usually inactivated by phosphorylation or inhibit something directly without phosphorylation.

We agree that there is no evidence that PD-180970 disrupts MTs by inhibiting phosphorylation of MAP70. In our live-imaging system, in which reagents are added to liquid cultivation medium, the time from the reagent application to the arrival at each cell varies. Therefore, in order to accurately measure the time required for the inhibitor to take effect, it is necessary to design a new assay system, such as using fluorescent dyes to monitor the reagent's diffusion. In addition, since some reactions mediated by protein phosphorylation occur rapidly, minute-order observations might not be sufficient. Therefore, as an alternative strategy to assess the direct involvement of MAP70 phosphorylation on MT stabilization, we will examine whether PD-180970 induces MT disruption using strains expressing the phospho-blocked (Ala) and phospho-mimic (Asp) versions of MAP70 described above.

3. Description of the revisions that have already been incorporated in the transferred manuscript

Reviewer #1:

Minor comments

The authors identified the analogs PD-166326 and PP1 as potent inhibitors of cell division. For completeness, it would be interesting to include a description of these arrest phenotypes and how they compare with that of PD180870 or PP2.

We have added the effects of all tested compounds on Arabidopsis embryos in Fig. S3C and Table S1. Based on this data and the results of tobacco BY-2 cells, we have compared the effects of PD-166326 and PD180870, and PP1 and PP2 in Results.

Although there are two more obvious candidates in the phosphoproteome datasets on which the authors focus on, there is very little discussion on whether the other top hits and whether they might be involved in cell division. On a related note, there is no discussion on the specificity of these compounds and the likelihood of phenotypes unrelated to cell division.

We have added the information of "Similar proteins in Arabidopsis" and "Description and putative functions" for all identified candidates for PD-180970 and PP2 in Table S2 and S3, respectively. With referring this information, we have added the sections to describe the possible contributions of these candidates on MT organization and phragmoplast formation in Results. In addition, we have described the specificity of these compounds and the phenotypes unrelated to cell division in the section for the results of Arabidopsis roots (Fig. S2A).

1st results section:

"...developed into the globular stage without causing morphological defects..."

Should omit the word "causing" or replace with "any/detectable"

We have omitted the word "causing".

Reviewer #2:

Even if the identification of the kinase(s) targeted by these two compounds is missing, the characterisation of at least two downstream effectors of these elusive kinase(s) inhibited by PD-180970 and PP2 is an important step forward. I would recommend to this point make very clear in the writing (e.g. already in the abstract). Upon a superficial reading, the reader could assume that MAP70s and PAKRP1s are the direct molecular targets of these compounds.

We appreciate the very positive comments. To clarify this point, in addition to the following responses to each suggestion, we have changed the last sentence of the abstract to "These properties make PD-180970 and PP2 useful tools for transiently controlling plant cell division at key manipulation nodes that are conserved in diverse plant species".

Revision Plan

Major comments

- I would modify the title to shift the emphasis from the methodology to the biological targets identified.

We have changed the title to "Identification of novel compounds inhibiting microtubule organization and phragmoplast formation in diverse plant species".

- Concerning MAP70s the authors claim that there is little functional data about this family. Yet, a recent paper (<https://www.science.org/doi/10.1126/sciadv.abm4974>) identifies MAP70-5 as necessary for the proper organisation of CMTs in the endodermis and its ability to actively remodel to accommodate emergence of the lateral root primordium in *Arabidopsis thaliana*. This could provide a functional context to test several of the predictions that the authors list in the discussion.

We have referred this paper in Results and Discussion, as "MAP70-5 was reported to increase MT length *in vitro* and to reorganize cortical MTs to alter the endodermal cell shape for lateral root initiation, suggesting that MAP70-5 mediates dynamic change of MT arrays".

Minor comments

- The narrative would be improved by moving the section "PD-180970 and PP2 do not irreversibly damage viability" before the phosphoproteomic section.

We have moved the "irreversibly" section to before the "phosphoproteomics" section.

Reviewer #3:

Minor comments

In supplemental data, authors show only 12 or 14 candidates of the target. It is interesting how other MAPs including homologues of MAP70 and Kinesin-12 in BY-2 cells were scored in the phospho-proteomics assay. I suggest authors show longer lists of proteomics including other MAPs. It would be valuable information for the research community.

We apologize for not providing the complete dataset. We have added Dataset S1 of total protein sequences that we predicted from published RNA-seq data of BY-2 cells, and all identified proteins of phosphoproteomics assay for PD-180970 and PP2 in Datasets S2 and S3, respectively. We have moved the lists of top candidates to Tables S2 and S3.

In Abstract, authors should mention that the two compounds reduced phosphorylation level of diverse proteins including MAP70 and Kinesin-12. This is very important results and, otherwise, it may cause misunderstanding of the activity of the compounds. In addition to this, it is better to rephrase the following sentence. "presumably by inhibiting MT-associated proteins (MAP70)" with "presumably by inhibiting phosphorylation of MT-associated proteins (MAP70)."

Revision Plan

To avoid such a misunderstanding, we have changed the descriptions in Abstract to "Phosphoproteomic analysis showed that these compounds reduced phosphorylation level of diverse proteins. In particular, PD-180970 inhibited phosphorylation of the conserved serine residues in MT-associated proteins (MAP70). PP2 significantly reduced the phosphorylation of class II Kinesin-12, and impaired its localization at the phragmoplast emerging site". Due to this change, the suggested sentence was eliminated. Also in Discussion, we have mentioned the reduction of phosphorylation of various proteins by stating, "we found that PD-180970 and PP2 reduced the phosphorylation levels of diverse proteins.

These parts may be further modified depending on the results of the phospho-blocked (Ala) and phospho-mimic (Asp) analyses.

Page7 line 1st. it would be better to insert "of MAP70 family" after "in the conserved MT-binding domain" because the MT binding domains are unique to the MAP70 family. I could not understand why this is " (2nd line) consistent with PD-18970 severely disrupting all the tested MT structure". At current stage, there is no evidence that dephosphorylation of MAP70 caused the microtubule disruption. I suggest authors remove the sentence (" , which was~MT structures").

We agreed on both points and have corrected them as the reviewer suggested.

4. Description of analyses that authors prefer not to carry out

None

August 8, 2022

Re: Life Science Alliance manuscript #LSA-2022-01657-T

Dr. Minako Ueda
Tohoku University
Graduate School of Life Sciences
Japan

Dear Dr. Ueda,

Thank you for submitting your manuscript entitled "Identification of novel compounds inhibiting microtubule organization and phragmoplast formation in diverse plant species" to Life Science Alliance. We invite you to re-submit the manuscript, revised according to your Revision Plan.

Thank you for this interesting contribution to Life Science Alliance. We are looking forward to receiving your revised manuscript.

Sincerely,

B. MANUSCRIPT ORGANIZATION AND FORMATTING:

Full Revision

Manuscript number: RC-2022-01469 (LSA-2022-01657-T)

Corresponding author(s): Minako, Ueda

1. General Statements

We really appreciate the positive comments and suggestions of the reviewers on our submitted manuscript. We think we have solved the issues inquired by reviewers, by adding new data and revising the phrases as detailed below.

Reviewer #1:

Major comments

Localization analysis of a transiently expressed MAP70 transgene with inactivating phosphosite mutations would be important to see whether the identified conserved phosphosites are relevant for MAP70 interaction with MTs. This experiment could be performed rapidly using transient expression in BY-2 cells.

Thank you for your suggestion. We fully agree with the importance of the proposed experiment. We performed transient expression of three types of fluorescent-tagged Nt-MAP70-2-like protein; unmodified (Ser), phospho-blocked (Ala) and phospho-mimic (Asp) in BY-2 cells harboring MT/histone marker. Unfortunately, MT organization itself was abolished in all cases, probably due to overexpressed effect of MAP70 protein (Fig S7). But the MAP70 signals were overlapped with MT signals, and thus we showed this result as "These results showed that highly expressed MAP70 can bind to MT and interfere with its organization regardless of the phosphorylation state of these three serines".

The authors propose that PP2 blocks phragmoplast formation by preventing phosphorylation of class II Kinesin-12 proteins. In support, authors show that PP2 treatment correlates with a decrease in KIN12A phosphopeptide count (not fully abolished) and its failure to localize to emerging phragmoplasts in BY-2 cells and Physcomitrium. As class II Kinesin-12 proteins have been previously implicated in phragmoplast assembly this is a fairly reasonable hypothesis, but would benefit from the analysis of transgenic KIN12A variants carrying inactivating (A) or potentially activating (D/E) phosphosite mutations. Is loss of phosphorylation sufficient to prevent phragmoplast localization? Can an activated variant rescue PP2-induced KIN12A localization and cell division defects? As above, using transient expression in BY-2 cells would be a fast approach to tackle these questions.

Full Revision

We agree that these experiments are crucial to understand the relation of PP2, KIN12A (PAKRP1), and phragmoplast formation. As performed with MAP70, we transiently expressed three versions of PAKRP1 (Ser/Ala/Asp) in BY-2 cell harboring MT/histone marker, and tested the effect of PP2. We found that PAKRP1 (Asp) localized on the phragmoplast, but it could not rescue PP2-induced defects of the protein localization and phragmoplast formation (Fig. S8B, and Movie S7). Therefore we showed this data as “This result suggests that either PAKRP1 (Asp) could not successfully mimic the constitutive phosphorylation, or that there are additional phosphorylation sites on PAKRP1 or other PP2 target proteins are involved”.

In addition, PAKRP1 (Ala) co-localized with entire MT, not specifically with phragmoplast, and it caused the disruption of MT pattern itself (Fig. S8C). Therefore we could not assess the direct relation between the loss of phosphorylation and the phragmoplast localization, and mentioned as “This suggests that highly expressed phospho-blocked PAKRP1 proteins were too much stabilized on MT, and thus disrupted MT organization.”.

Since our additional experiments did not provide evidence that PD-180970 and PP2 act by altering the phosphorylation of MAP70 and Kinesin-12, respectively, we have weakened this statement overall in the current manuscript. We have modified our hypothetical model (Fig 4C) to add the potential effects of PD-180970 and PP2 not via MAP70 and Kinesin-12, respectively.

Minor comments

The authors identified the analogs PD-166326 and PP1 as potent inhibitors of cell division. For completeness, it would be interesting to include a description of these arrest phenotypes and how they compare with that of PD180870 or PP2.

We have added the effects of all tested compounds on Arabidopsis embryos in Fig. S3C and Table S1. Based on this data and the results of tobacco BY-2 cells, we have compared the effects of PD-166326 and PD180970, and PP1 and PP2 in Results.

Although there are two more obvious candidates in the phosphoproteome datasets on which the authors focus on, there is very little discussion on whether the other top hits and whether they might be involved in cell division. On a related note, there is no discussion on the specificity of these compounds and the likelihood of phenotypes unrelated to cell division.

We have added the information of “Similar proteins in Arabidopsis” and “Description and putative functions” for all identified candidates for PD-180970 and PP2 in Table S2 and S3, respectively. With referring this information, we have added the sections to describe the possible contributions of these candidates on MT organization and phragmoplast formation in Results. In addition, we have described the specificity of these compounds and the phenotypes unrelated to cell division in the section for the results of Arabidopsis roots (Fig. S2A).

1st results section:

“...developed into the globular stage without causing morphological defects...”

Should omit the word "causing" or replace with "any/detectable"

Full Revision

We have omitted the word "causing".

Reviewer #2:

Even if the identification of the kinase(s) targeted by these two compounds is missing, the characterisation of at least two downstream effectors of these elusive kinase(s) inhibited by PD-180970 and PP2 is an important step forward. I would recommend to this point make very clear in the writing (e.g. already in the abstract). Upon a superficial reading, the reader could assume that MAP70s and PAKRP1s are the direct molecular targets of these compounds.

We appreciate the very positive comments. To clarify this point, we have toned down the mention of MAP70s and PAKRP1s in the abstract to "Phosphoproteomic analysis showed that these compounds reduced the phosphorylation of diverse proteins including MT-associated proteins (MAP70) and class II Kinesin-12." We also have changed the last sentence of the abstract to "These properties make PD-180970 and PP2 useful tools for transiently controlling plant cell division at key manipulation nodes conserved across diverse plant species".

Major comments

- I would modify the title to shift the emphasis from the methodology to the biological targets identified.

We have changed the title to "Novel inhibitors of microtubule organization and phragmoplast formation in diverse plant species".

- Concerning MAP70s the authors claim that there is little functional data about this family. Yet, a recent paper (<https://www.science.org/doi/10.1126/sciadv.abm4974>) identifies MAP70-5 as necessary for the proper organisation of CMTs in the endodermis and its ability to actively remodel to accommodate emergence of the lateral root primordium in *Arabidopsis thaliana*. This could provide a functional context to test several of the predictions that the authors list in the discussion.

We appreciate for providing information. We have referred this paper in Discussion, as "MAP70-5 was reported to increase MT length *in vitro* and to reorganize cortical MTs to alter the endodermal cell shape for lateral root initiation, suggesting that MAP70-5 mediates dynamic change of MT arrays".

- The manuscript would strongly benefit from being revised by a native english speaker. There are many unusual or awkward formulation, in particular in the abstract.

Full Revision

We apologize for unnatural sentences. After adding new data and correcting the manuscript, we have asked a native English speaker to revise it.

Minor comments

- The narrative would be improved by moving the section "PD-180970 and PP2 do not irreversibly damage viability" before the phosphoproteomic section.

We have moved the "irreversibly" section to before the "phosphoproteomics" section.

Reviewer #3:

Major comments

The major concern is lack of evidence to connect MAP70 and MT disruption upon treatment with PD-180970, in contrast to PP2, which was shown to affect localization of Kinesin-12. I wonder if authors could use taxol to stabilize MTs, then observe the localization of MAP70 with application of PD-180970?

We totally agree with your concern. As we responded to Reviewer 1, we temporally expressed fluorescent-tagged MAP70 in BY-2 cells harboring MT/histone marker. Unfortunately, even in this transient system, MAP70 caused severe disruptions in MT patterns (Fig. S7B), so we declined to further evaluate the effect of PD-180970. Because taxol was not a very effective inhibitor in our system, we instead tested phosphosite mutations to assess the relationship of MAP70, MT and PD-180970. But both of phospho-blocked (Ala) and phospho-mimic (Asp) caused same MT disorganization as found with unmodified MAP70 (Fig. S7C and D). Therefore we could not confirm their relationship, but all MAP70 versions showed co-localization with MT signal, and thus we mentioned as "These results showed that highly expressed MAP70 could bind to MT and interfere with its organization".

I have another concern on the action of PD-180970. PD-180970 appears to affect ubiquitously indispensable proteins for MTs. If PD-180970 disrupts MT by inhibiting phosphorylation of some MAPs, it must need time for turnover of proteins phosphorylated before PD-180970 was applied. In the proteomics experiment, authors treated the cells with the compounds for 8-9 hr. On the other hand, in BY-2 cells, PD-180970 disrupted MTs only 30 min after application of PD-180970. I wonder if proteins were replaced during the 30 min. Could authors examine how long it takes to affect interphase MTs? If PD-180970 disrupts MTs in a 5-10 min like oryzalin, it is unlikely that inhibition of phosphorylation of proteins like MAP70 caused MT disruption. Rather, it may inhibit some proteins that have activity to disrupt microtubules but are usually inactivated by phosphorylation or inhibit something directly without phosphorylation.

We totally agree that we don't have any evidence that PD-180970 disrupts MTs by inhibiting MAP70 phosphorylation. In our live-imaging system, in which reagents are added to liquid

Full Revision

cultivation medium, is not suitable for accurately measuring the time required for the inhibitor to take effect because of the time variation from the reagent application until it reaches individual cells. Therefore, as an alternative strategy to assess the direct involvement of MAP70 phosphorylation on MT stabilization, we examined the effect of phosphorylation site mutations. As mentioned above, however, all versions disrupted MT pattern even without PD-180970. Therefore we could not investigate the relevance of MAP70 phosphorylation and MT organization, but this result might support the possibility that MAP70 has the activity to disrupt MTs, and that PD-180970 stabilizes MAP70 on MTs not via phosphorylation and thus disorganize MTs. We have added this point in Discussion as "we cannot exclude the possibility that PD-180970 disrupts MT organization by stabilizing MAP70 (and other proteins) on MT without affecting phosphorylation since MAP70 overexpression caused MT disorganization regardless of the phosphorylation state and PD-180970 rapidly disrupted MTs only 30 min after application (Figs. 3, 4, and S7).".

Minor comments

In supplemental data, authors show only 12 or 14 candidates of the target. It is interesting how other MAPs including homologues of MAP70 and Kinesin-12 in BY-2 cells were scored in the phospho-proteomics assay. I suggest authors show longer lists of proteomics including other MAPs. It would be valuable information for the research community.

We apologize for not providing the complete dataset. We have added Dataset S1 of total protein sequences that we predicted from published RNA-seq data of BY-2 cells, and all identified proteins of phosphoproteomics assay for PD-180970 and PP2 in Datasets S2 and S3, respectively. We have moved the lists of top candidates to Tables S2 and S3.

In Abstract, authors should mention that the two compounds reduced phosphorylation level of diverse proteins including MAP70 and Kinesin-12. This is very important results and, otherwise, it may cause misunderstanding of the activity of the compounds. In addition to this, it is better to rephrase the following sentence. "presumably by inhibiting MT-associated proteins (MAP70)" with "presumably by inhibiting phosphorylation of MT-associated proteins (MAP70)."

We appreciate your suggestion, and have changed the descriptions in Abstract to "Phosphoproteomic analysis showed that these compounds reduced the phosphorylation of diverse proteins, including MT-associated proteins (MAP70) and class II Kinesin-12". As answered to Reviewer #2, the description "presumably by inhibiting MT-associated proteins (MAP70)" itself has been deleted from the current Abstract.

Page7 line 1st. it would be better to insert "of MAP70 family" after "in the conserved MT-binding domain" because the MT binding domains are unique to the MAP70 family. I could not understand why this is " (2nd line) consistent with PD-18970 severely disrupting all the tested MT structure". At current stage, there is no evidence that dephosphorylation of MAP70 caused

Full Revision

the microtubule disruption. I suggest authors remove the sentence (" , which was~MT structures").

We agreed on both points and have corrected them as the reviewer suggested.

February 13, 2023

RE: Life Science Alliance Manuscript #LSA-2022-01657-TR

Dr. Minako Ueda
Tohoku University
Graduate School of Life Sciences
6-3, Aramaki Aza-Aoba, Aoba-ku
Sendai 980-8578
Japan

Dear Dr. Ueda,

Thank you for submitting your revised manuscript entitled "Novel inhibitors of microtubule organization and phragmoplast formation in diverse plant species". We would be happy to publish your paper in Life Science Alliance pending final revisions necessary to meet our formatting guidelines.

- please address Reviewer 2 and 3's remaining comments
- please add the Twitter handle of your host institute/organization as well as your own or/and one of the authors in our system
- please incorporate your supplementary methods into your main materials and methods section; we do not have a word limit for this section. Same with incorporating the Supplementary References into the main Reference list.
- please add your supplementary figure legends and your video legends to the main manuscript text

A. FINAL FILES:

B. MANUSCRIPT ORGANIZATION AND FORMATTING:

**Submission of a paper that does not conform to Life Science Alliance guidelines will delay the acceptance of your

manuscript.**

The license to publish form must be signed before your manuscript can be sent to production. A link to the electronic license to publish form will be sent to the corresponding author only. Please take a moment to check your funder requirements.

Sincerely,

Reviewer #1 (Comments to the Authors (Required)):

the authors have addressed all the point I raised in this revised manuscript.

Reviewer #2 (Comments to the Authors (Required)):

I found that authors added appropriate changes to the manuscript. I really thank authors for providing the sequence data set table S1. The abstract and discussion were nicely edited so that the manuscript well reflects the present situation of the study.

Although the detailed mechanisms by which the identified compounds affect cell division and microtubule organization are still to be determined, I believe that the compounds will be very useful for exploring the mechanisms underlying plant cell division and microtubule organization.

Minor

In the legend of Supplementary Figure7, to clarify the identity of the observed MAP70, it would be better to rephrase "mClover-MAP70" as "mClover-Nt-MAP70-2-like (MAP70). Similarly, in the legend and text for Supplementary Figure8, please indicate At or Nt for the observed kinesin.

Reviewer #3 (Comments to the Authors (Required)):

In their paper, the authors use a very elaborate and laborious screening on Arabidopsis early embryos to identify compounds that can change cell division (and shape). In doing so, they identify such compounds and show that their effects are generic, both with respect to developmental stage, cell type, and organism. Using a biochemical approach, they identify potential target proteins. Validation of these is still open to (some) debate, but the work presents a very important advance and a new set of chemical tools for plant cell biology.

I am in general very supportive of publication. As a reviewer of the initial version (for Review Commons), I made several comments, which have been addressed well. My only comments relate to the description and interpretation of the characterization of the drug target proteins. Here, some more caution would be warranted. For example, I am not convinced that one can "deem these proteins targets of the compounds", as the section titles suggest.

Response to reviewer #2

Comments to the Authors (Required):

I found that authors added appropriate changes to the manuscript. I really thank authors for providing the sequence data set table S1. The abstract and discussion were nicely edited so that the manuscript well reflects the present situation of the study.

Although the detailed mechanisms by which the identified compounds affect cell division and microtubule organization are still to be determined, I believe that the compounds will be very useful for exploring the mechanisms underlying plant cell division and microtubule organization.

We really appreciate for the positive comments.

Minor

In the legend of Supplementary Figure7, to clarify the identity of the observed MAP70, it would be better to rephrase "mClover-MAP70" as "mClover-Nt-MAP70-2-like (MAP70)". Similarly, in the legend and text for Supplementary Figure8, please indicate At or Nt for the observed kinesin.

We agree with this point, and we have specified At or Nt for all descriptions of MAP70 and kinesin.

Response to reviewer #3

Comments to the Authors (Required):

In their paper, the authors use a very elaborate and laborious screening on Arabidopsis early embryos to identify compounds that can change cell division (and shape). In doing so, they identify such compounds and show that their effects are generic, both with respect to developmental stage, cell type, and organism. Using a biochemical approach, they identify potential target proteins. Validation of these is still open to (some) debate, but the work presents a very important advance and a new set of chemical tools for plant cell biology.

I am in general very supportive of publication. As a reviewer of the initial version (for Review Commons), I made several comments, which have been addressed well. My only comments relate to the description and interpretation of the characterization of the drug target proteins. Here, some more caution would be warranted. For example, I am not convinced that one can "deem these proteins targets of the compounds", as the section titles suggest.

We really appreciate for the positive comments, and agree to your concern about "target proteins". Therefore, we have toned down our statements and changed the section titles from "MAP70 deemed a candidate of PD-180970" to "Potential downstream targets for PD-180970" and "Class II Kinesin-12 deemed a candidate of PP2 targets" to "Potential downstream targets for PP2", respectively. Also, to clarify the indirect inhibition, we changed the statements of "targets" into "downstream targets" or "substrates" throughout the manuscript.

February 14, 2023

RE: Life Science Alliance Manuscript #LSA-2022-01657-TRR

Dr. Minako Ueda
Tohoku University
Graduate School of Life Sciences
6-3, Aramaki Aza-Aoba, Aoba-ku
Sendai 980-8578
Japan

Dear Dr. Ueda,

Thank you for submitting your Research Article entitled "Novel inhibitors of microtubule organization and phragmoplast formation in diverse plant species". It is a pleasure to let you know that your manuscript is now accepted for publication in Life Science Alliance. Congratulations on this interesting work.

DISTRIBUTION OF MATERIALS:

Again, congratulations on a very nice paper. I hope you found the review process to be constructive and are pleased with how the manuscript was handled editorially. We look forward to future exciting submissions from your lab.

Sincerely,
